# Evaluation of the Antioxidant and Antiradical Properties of Some Phyto and Mammalian Lignans

**DOI:** 10.3390/molecules26237099

**Published:** 2021-11-24

**Authors:** Leyla Polat Kose, İlhami Gulcin

**Affiliations:** 1Department of Pharmacy Services, Vocational School, Beykent University, Istanbul 34500, Turkey; lylpolat@atauni.edu.tr; 2Department of Chemistry, Faculty of Science, Ataturk University, Erzurum 25240, Turkey

**Keywords:** phenolic compounds, lignan, antioxidant activity, antiradical activity, reducing power

## Abstract

In this study, the antioxidant and antiradical properties of some phyto lignans (nordihydroguaiaretic acid, secoisolariciresinol, secoisolariciresinol diglycoside, and α-(-)-conidendrin) and mammalian lignans (enterodiol and enterolactone) were examined by different antioxidant assays. For this purpose, radical scavenging activities of phyto and mammalian lignans were realized by 2,2′-azino-bis (3-ethylbenzothiazoline-6-sulphonic acid) radical (ABTS^•+^) scavenging assay and 1,1-diphenyl-2-picrylhydrazyl radical (DPPH) scavenging assay. Additionally, the reducing ability of phyto and mammalian lignans were evaluated by cupric ions (Cu^2+^) reducing (CUPRAC) ability, and ferric ions (Fe^3+^) and [Fe^3+^-(TPTZ)2]^3+^ complex reducing (FRAP) abilities. Also, half maximal inhibitory concentration (IC_50_) values were determined and reported for DPPH^•^ and ABTS^•+^ scavenging influences of all of the lignan molecules. The absorbances of the lignans were found in the range of 0.150–2.320 for Fe^3+^ reducing, in the range of 0.040–2.090 for Cu^2+^ reducing, and in the range of 0.360–1.810 for the FRAP assay. On the other hand, the IC_50_ values of phyto and mammalian lignans were determined in the ranges of 6.601–932.167 µg/mL for DPPH^•^ scavenging and 13.007–27.829 µg/mL for ABTS^•+^ scavenging. In all of the used bioanalytical methods, phyto lignans, as secondary metabolites in plants, demonstrated considerably higher antioxidant activity compared to that of mammalian lignans. In addition, it was observed that enterodiol and enterolactone exhibited relatively weaker antioxidant activities when compared to phyto lignans or standard antioxidants, including butylated hydroxytoluene (BHT), butylated hydroxyanisole (BHA), Trolox, and α-tocopherol.

## 1. Introduction

Phytolignans are found in high amounts in flaxseed, cereal, whole wheat, and vegetables; in some fruits such as cherries, strawberries; and tea. In addition, they exist in urine, plasma, saliva, sperm, and prostatic fluids in the human body [1]. Phytoestrogens are separated into different subclasses according to their sources. They are isoflavones, kumestans, lignans, and stilbenes. Lignans are a main class of secondary metabolites that occur in two phenylpropanoid molecules by undergoing oxidative dimerization in plants [2]. These phenolic molecules are found in the structure of many plants used in Eastern medicine. According to recent studies, lignans provide very important biological and pharmacological properties in plants [3]. Although lignans are mostly found in free form in nature, a few of them are also found in the form of glycoside derivatives. They are found in different parts of plants such as leaves, roots, seeds, stems, and fruits. Despite their widespread distribution in the plant kingdom, their biological functions in plants are still not fully determined. Thanks to the antiviral, antimicrobial, antioxidant, antifungal, and insecticidal properties of some lignans, these molecules have a strong effect on some biological pathogens. Moreover, these phenolic compounds can contribute to both the growth and development of plants [4]. It was reported that lignans had a wide range of biological activity, including immunosuppressive, anti-inflammatory, cardiovascular, antioxidant, antitumor, and antiviral effects [2,5].

Lignans protect cells from oxidative damage caused by reactive oxygen species (ROS) by helping to increase the amount of glutathione, which acts as an antioxidant in tissues [6]. As previously mentioned, phyto lignan sources undergo deglycosylation after being taken in the diet, and some are converted to mammalian lignans, also known as enterolignans, by colon bacteria [7]. Enterodiol and enterolactone (Figure 1), known as enterolignans, exhibit antitumor, weak estrogenic, antiestrogenic, and antioxidant properties. They are found in many body fluids, such as plasma, saliva, and prostatic fluid [8].

Lignans also have effects on enzymes, angiogenesis, proliferation, cell differentiation, protein synthesis, and growth factor. In human beings, plant lignans are deglycolysed after digestion, then transformed by microorganisms into mammalian lignans, namely enterolactone and enterodiol, which are known enterolignans [2]. Although lignans are common in plants, they are rarely studied because they are difficult to isolate, purify, and analyze.

Normally, the electrons in atoms are located opposite from each other in pairs in areas called orbitals. As a result of the interactions between atoms, chemical bonds are formed, and the molecular structure is formed in the transition from the atomic structure. Single electron structures in atomic or molecular structures are called free radicals. Free radicals are highly energetic and unstable compounds containing unpaired electrons in their external orbitals [9,10]. Free radicals are unstable because they exchange electrons very easily and quickly with other atoms or molecules. Free radicals are also called oxidant molecules or ROS [11,12,13]. Various ROS can form in different ways in living organisms [14,15]. In aerobic organisms, ROS are always encountered during oxidative phosphorylation and in biologically important metabolic pathways such as active phagocytosis [16,17]. The main ROS that occurs during the formation of these metabolic events are as follows: hydroxy radical (HO^•^), peroxy radical (ROO^•^), hydroperoxy radical (HOO^•^), alkoxy radical (RO^•^), and superoxide anion (O_2_^•−^). In addition, there are non-free-radical oxygen species such as hydrogen peroxide (H_2_O_2_), peroxynitrite (ONOO^−^), singlet oxygen (^1^O_2_), and hypochlorous acid (HClO) [18,19,20,21]. Free radicals can be formed by factors both inside and outside the body. One of the mechanisms in the immune system to neutralize bacteria and viruses in the body is the formation of free radicals. Free radicals are also produced by biological events in the human body, such as oxygenated breathing, metabolism, fatty acid oxidation, immunological reactions, and infection [22,23]. In normal cells, there is a balance between oxidant and antioxidant levels. However, the state of balance can be adversely altered when the production of ROS is increased or the antioxidant levels are decreased in the body. In the case of a high ROS level, it can damage to many molecules, including proteins, lipids, RNA, and DNA due to their high reactivity. This condition is called oxidative stress [24,25,26].

Antioxidants are compounds that protect the living system from harmful effects of free radicals and ROS. These biological active compounds are effective even at very low concentrations [27]. They break the auto-oxidative chain reaction initiated by ROS and neutralize the damages of free radicals and ROS from industrial pollution, radiation, and unhealthy eating habits. The most commonly used antioxidants are butylated hydroxytoluene (BHT), butylated hydroxyanisole (BHA), tert-butylhydroquinone, and propyl gallate. However, they are suspected of being responsible for carcinogenesis and liver damage [28]. So, people prefer safer and natural antioxidants with plant origins [29]. Natural antioxidants are classified as either antioxidants synthesized by the organism (endogen) or taken with nutrients from the outside (exogen). However, the synthesis of antioxidants in living systems also decreases with ageing. Researches have proved that plant-based antioxidants are a good alternative to overcome this situation [30,31]. Fruits and vegetables, which are some of the most important sources of herbal antioxidants, play an important role in eliminating the damage of ROS, preventing abnormal cell growth and repairing damaged cells [32]. The most basic principle of antioxidants’ activity is that they have electrons to neutralize free radicals in the medium [33]. In addition, antioxidant activity of a molecule may differ depending on number and position of the hydroxyl (-OH) groups in the aromatic ring [9]. In particular, the addition of an electron-donating group, such as a -OH group in the *ortho*- or *para*-position of a phenolic molecule, increases the antioxidant activity of a phenol or phenolic acid [10].

In this study, the antioxidant and antiradical properties of some phyto and mammalian lignans were determined at various concentrations using different bioanalytical and antioxidant methods, including Fe^3+^ reducing ability, Fe^3+^-TPTZ reduction capacity, Cu^2+^ reduction ability, and ABTS^•+^ and DPPH^•^ radical scavenging activities. Their antioxidant potentials also were compared to some putative and standard antioxidants, including BHA, BHT, α-tocopherol, and Trolox.

## 2. Results

There are numerous antioxidant assays and modifications thereof for evaluation of antioxidant activity in pure compounds. Of these, we selected a few of the most outstanding of these methods. To this end, the reducing capabilities of phyto and mammalian lignans were realized by three distinct and different reducing-power methods, including Fe^3+^(CN-)_6_ reduction, Fe^3+^-TPTZ reduction, and Cu^2+^ ion reduction capacities. In addition, these reduction abilities were also tested and compared with BHA, BHT, α-tocopherol and Trolox as standard molecules. According to the obtained data, it was determined that the phyto and mammalian lignans had very good reducing powers, as expected, depending on their chemical structures. In this context, at the same concentration (10 µg/mL), the reducing abilities of phyto lignans, mammalian lignans, and standards were found in the following order: nordihydroguaiaretic acid (λ_700_: 2.320; r^2^: 0.7704) > BHA (λ_700_: 1.480; r^2^: 0.9505) > (-)-secoisolariciresinol (λ_700_: 1.310; r^2^: 0.9655) > BHT (λ_700_: 0.990; r^2^: 0.9198) > secoisolariciresinol diglycoside (λ_700_: 0.930; r^2^: 0.9425) > α-(-)-conidendrin (λ_700_: 0.760; r^2^: 0.9086) > α-tocopherol (λ_700_: 0.540; r^2^: 0.9841) > Trolox (λ_700_: 0.520; r^2^: 0.9907) > enterodiol (λ_700_: 0.160; r^2^: 0.6270) > enterolactone (λ_700_: 0.150; r^2^: 0.6511) (Figure 2A and Table 1). With the results obtained, it was found that the lignans had a good ability to reduce iron ions (Fe^3+^) in the medium, and had the ability to generate stable products by giving electrons to free radicals and ROS. For this reason, the color of the solution medium turned from yellow to green in the presence of phyto and mammalian lignans.

Cu^2+^ reducing capability of phyto and mammalian lignans and positive controls at the same concentration (10 μg/mL) showed the following order: nordihydroguaiaretic acid (λ_450_: 2.090; r^2^: 0.4703) > BHA (λ_450_: 1.320; r^2^: 0.8290) > secoisolariciresinol (λ_450_: 1.180; r^2^: 0.8665) > BHT (λ_450_: 1.000; r^2^: 0.8589) > α-(-)-conidendrin (λ_450_: 0.670; r^2^: 0.9666) > (-)-secoisolariciresinol diglycoside (λ_450_: 0.590; r^2^: 0.9917) > Trolox (λ_450_: 0.530; r^2^: 0.9799) > α-tocopherol (λ_450_: 0.460; r^2^: 0.9971) > enterodiol (λ_450_: 0.060; r^2^: 0.8867) > enterolactone (λ_450_: 0.040; r^2^: 0.7938) (Figure 2B and Table 1). This method is easy, fast, selective, and inexpensive. It also can be easily applied to many different antioxidant systems. The CUPRAC method is substantially a chromogenic redox reaction that occurs at around neutral pH and can be completed within half an hour [34].

As can be seen in Figure 2C and Table 1, the TPTZ-Fe^3+^ complex reducing ability of phyto and mammalian lignans and standards at a concentration of 10 μg/mL showed the following order: nordihydroguaiaretic acid (λ_593_:1.810; r^2^: 0.5289) > (-)-secoisolariciresinol (λ_593_: 1.770; r^2^: 0.5248) > BHA (λ_593_: 1.560; r^2^: 0.6352) > α-(-)-conidendrin (λ_593_: 1.320; r^2^: 0.7906) > secoisolariciresinol diglycoside (λ_593_: 1.170; r^2^: 0.8301) > Trolox (λ_593_:1.130; r^2^: 0.8600) > BHT (λ_593_: 1.010; r^2^: 0.9085) > α-tocopherol (λ_593_: 0.670; r^2^: 0.9815) > enterolactone (λ_593_: 0.370; r^2^: 0.4823) > enterodiol (λ_593_: 0.360; r^2^: 0.4483) (Figure 2C and Table 1). During the evaluation of the reducing capacity of lignan molecules, the FRAP method was preferred due to some advantages.

Assays based on the use of DPPH and ABTS radicals are the most popular spectrophotometric methods for determining the antioxidant capacity of pure molecules such as lignans. Figure 3A and Table 2 show that a crucial concentration decreasing (*p* < 0.01) of the DPPH radical thanks to the scavenging capability of lignan molecules and standards. IC_50_ values were found as 6.601 µg/mL for nordihydroguaiaretic acid (r^2^: 0.8498) > 6.478 µg/mL for Trolox (r^2^: 0.9624) > 8.864 µg/mL for α-tocopherol (r^2^: 0.9437) > 8.886 µg/mL for BHA (r^2^: 0.8406) > 14.141 µg/mL for (-)-secoisolariciresinol (r^2^: 0.8426) > 16.970 µg/mL for secoisolariciresinol diglycoside (r^2^: 0.8998) > 17.778 µg/mL for BHT (r^2^: 0.9397) > 23.296 µg/mL for α-(-)-conidendrin (r^2^: 0.9281) > 770.164 µg/mL for enterodiol (r^2^: 0.7746) > 932.167 µg/mL for enterolactone (r^2^: 0.9792). A low IC_50_ value indicated a high DPPH radical scavenging effect. Another method used as a radical scavenging method is the ABTS^•+^ scavenging method [35].

As seen in Figure 3B and Table 2, some phyto and mammalian lignans showed concentration-dependent (10–30 µg/mL) scavenging effects against ABTS radicals. The IC_50_ values for lignan molecules in this analysis were found as 12.252 µg/mL for (-)-secoisolariciresinol (r^2^: 0.8001) > 13.007 µg/mL for BHT (r^2^: 0.9116) > 13.070 µg/mL for nordihydroguaiaretic acid (r^2^: 0.9174) > 13.345 µg/mL for α-(-)-conidendrin (r^2^: 0.9140) > 13.547 µg/mL for (-)-secoisolariciresinol diglycoside (r^2^: 0.9467) > 13.378 µg/mL for enterodiol (r^2^: 0.8163) > 14.146 µg/mL for enterolactone (r^2^: 0.9070) > 14.264 µg/mL for Trolox (r^2^: 0.9660) > 16.552 µg/mL for BHA (r^2^: 0.9837) > 27.829 µg/mL for α-tocopherol (r^2^: 0.9816). With the addition of some phyto and mammalian lignans, a significant decrease in ABTS radicals occurred in the reaction medium. In this experiment, it was determined that some phyto and mammalian lignans with low IC_50_ values obtained numerically had a strong ABTS^•+^ scavenging activity.

## 3. Discussion

The antioxidant efficiency of a pure compound can occur through different mechanisms, such as transition metal ion chelation, cleavage of peroxides, inhibition of hydrogen abstraction, and radical elimination [36]. For example, in a system in which oxidation is accelerated by transition metals, the reduction of antioxidant properties of a compound is not significant. However, even if the antioxidant molecule only consists of the metal-chelating ability, it will stop or slow down the oxidation in such a system [37]. Additionally, electron-withdrawing capacity reflects the reducing power of a compound [38]. Antioxidants may be in the form of stabilizing oxidants in reductant and redox reactions. The reduction capacity of an antioxidant molecule can be recorded by different methods [39]. Recently, there has been an increased interest in phytochemical-derived food, which is known to be beneficial for human health [40,41].

Edible plants contain biphenolic compounds called hormonelike phytoestrogens. They are divided into two main groups, including isoflavones and lignans. Isoflavones are a broad and very different flavonoid subclass. Lignans are defined by the 2,3-dibenzylbutane skeleton and are found in all veiny plants. They are present in high concentrations in seeds, fruits, vegetables, and beverages made from these. The basic lignans found in plants are matairesinol, secoisolariciresinol, lariciresinol, and pinoresinol. It was reported that phyto lignans had better antioxidant activity when compared to mammalian lignans, including enterodiol and enterolactone. They produce enterolactone and enterodiol, which are two basic compounds with estrogenic activity in humans [42]. In addition, lignans show important impacts on enzyme activities, protein synthesis, proliferation, angiogenesis, growth factor, and cell differentiation [43]. As known, free radicals and ROS can be formed in living organisms, and are responsible for many neurological diseases such as Parkinson’s disease, Alzheimer’s disease, and especially cancer types [42]. For these reasons, antioxidant systems are constantly needed in organisms. Therefore, antioxidant supplements are of great interest today. Antioxidant compounds that are taken up with natural products or isolated from natural products are a correct alternative in the fight against diseases caused by ROS species [44,45,46]. Phytolignans as secondary metabolites are widely distributed in grains, fruits, cereals, and vegetables [47]. They are bioactive compounds with a wide scale of biological activities including antitumor, antioxidant, and weakly estrogen-linked properties. There is great interest in promoting the inclusion of phyto lignan-rich foods in human diets due to their potential human health benefits, including the prevention of hypercholesterolemia, cardiovascular disease, breast and prostate cancers, osteoporosis, and menopausal symptoms [48]. It was reported that humans rely on gut microbes to convert plant lignans into mammalian lignans. Intestinal bacteria metabolize plant lignans by processes such as glycosylation, demethylation, methylenation, dehydroxylation, ring cleavage, and oxidation [49]. Several human intestinal bacteria involved in these processes have been isolated and characterized. In addition, it is known that phytolignans are metabolized to mammalian lignans by both fecal and ruminal microbiota (Figure 4). Recent studies have shown that lignan metabolism in ruminant animals mainly occurs in the rumen, where plant lignans are converted to mammalian lignans by rumen microbes [50]. Plant lignans are converted to mammalian lignans by the gut microbiota and have three metabolic fates. First, they are excreted directly with the feces. Second, they are absorbed by the epithelial cells lining the colon, and are conjugated with glucuronic acid or sulfate. Third, they can be absorbed from the gut in their deconjugated form, and released into the bloodstream as conjugated free forms and reach the liver. Eventually, the conjugated mammalian lignans are excreted into physiological fluids, including plasma and urine, or returned to the gut via enterohepatic circulation [49].

Although lignans and their biogenetic precursors have various health benefits, their application as antioxidants in the food industry is limited due to the poor liposolubility of such phenolic compounds. However, due to the positive and potential health benefits of phyto lignans, their usage as functional foods or as beneficial food additives is of great interest [47]. The antioxidant abilities of secoisolariciresinol diglycoside as a glycosylated lignan have been recently studied. In addition, secoisolariciresinol diglycoside is deglycosylated by microorganisms in the gut and forms secoisolariciresinol, which, in turn, is converted into enterolactone (Figure 4). The presence of two hydrophilic glucose residues along with two phenol groups in the structure of secoisolariciresinol diglycoside is certain to reduce its liposolubility. The use of natural antioxidants such as lignans in the food industry is an important issue to consider. It is very important that they are dissolved in lipid systems. Where these properties are insufficient and chemical modifications can sometimes solve problems.

In this study, the antioxidant activities of some phyto and mammalian lignans were determined using several distinct antioxidant methods supporting each other. The first used method with this purpose was Fe^3+^ reduction ability. Generally, the reduction properties of a compound depend on the existence of reducing agents with a hydrogen atom donor [51,52]. Thanks to the Fe^3+^(CN^−^)_6_ reducing assay, reducing abilities of some phyto and mammalian lignans were defined. The Fe^3+^(CN^−^)_6_ and Cu^2+^ ion reducing abilities of phyto and mammalian lignans were also tested and compared with BHA, BHT, α-tocopherol and Trolox as standard antioxidants.

According to the obtained data, it also was determined that the phyto and mammalian lignans had very good reducing power, as expected, depending on their chemical structures. The ferric reducing antioxidant power (FRAP) assay is a simple test that uses a reductant in a redox reaction with an oxidant and is combined with a colorimetric method [53]. The addition of Fe^3+^ to the reduced product by addition of phyto and mammalian lignans leads to the formation of Fe_4_[Fe(CN)_6_], a complex in the Prussian blue color with sharp absorbance at 700 nm [54,55].

Additionally, the absorbance values for Cu^2+^-reducing ability had been reported for some natural products obtained from plants as 0.277 (r^2^: 0.9836) for usnic acid [30], 0.762 for eugenol (r^2^: 0.9957) [56], 1.085 (r^2^: 0.8403) for resveratrol [57], 0.750 (r^2^: 0.9550) for taxifolin [58], and 1.314 (r^2^: 0.9682) for olivetol [59] at the same concentration. A relationship was observed between the three reduction methods applied to define the reduction capacities of phyto and mammalian lignans in the studies conducted. In addition, phyto and mammalian lignans and the standards demonstrated similar activities in all reduction methods [60]. This was due to the presence of aromatic -OH groups in the phyto and mammalian lignans, which could be reduced. Delocalization caused by the resonance structures of the electrons in the aromatic ring increased the acidity of the aromatic -OH groups, resulting in a stronger reducing agent [61,62]. The Cu^2+^ reduction abilities of phyto and mammalian lignans and standards are shown in Figure 2B. As a result of the studies, it was determined that there was a suitable correlation between the reduction power of Cu^2+^ and the increasing concentrations of phyto and mammalian lignans. The numerical data of the Cu^2+^ reducing power, which increased as the concentrations of phyto and mammalian lignans increased, are shown in Table 1 (10 μg/mL).

The FRAP method can be used for detection of the total reducing ability of both pure antioxidant molecules and plant extracts [63]. The FRAP method is based on the principle that the phenolic substances used shown antioxidant activity by donating electrons [64]. In this method, ferric ions (Fe^3+^) are reduced to ferrous ions (Fe^2+^). This ion formed in the environment after interaction with reducing molecules, forming a dark blue complex with tripyridyl triazine (TPTZ). The colored complex (TPTZ-Fe^2+^) was formed after the reduction reaction and showed a maximum absorbance at 593 nm [65]. In this study, which we conducted on the basis of the FRAP method, it was determined that as the concentrations of phyto and mammalian lignans we used increased, the reducing capacity of Fe^3+^ ions to Fe^2+^ ions increased strongly. Primarily, the FRAP method is comparatively basic and can be standardized easily and in a short time [66]. In addition, this reduction assay is a widely used method for rapid identification of antioxidant capacities of various foods, phenolic compounds, and pharmaceutical plants. Furthermore, it has been quite practical for detecting antioxidant potential of phenolics such as polyphenols and flavonoids as in vitro systems [67].

The radical scavenging activities of antioxidant compounds are very significant in terms of damage to the organism by free radicals in biological system, food, and pharmaceutical applications [68,69]. In addition, ABTS^•+^ and DPPH^•^ scavenging methods are fast, simple, selective, and repeatable. Therefore, they are widely used for definition of radical scavenging activities of pure substances. It is easy to use violet DPPH∙ and green-blue ABTS^•+^ chromogens, which have high sensitivity [70]. The DPPH method is commonly used to diagnose that antioxidant molecules destroy free radicals. For the DPPH· removing capacity, the Blois method was firstly used [71]. In this method, the antioxidant agent that is a hydrogen donor is mixed efficiently, scavenging the DPPH radicals in the alcoholic medium [72,73]. This study clearly showed that lignans, especially phyto lignans, strongly scavenged DPPH radicals and sharply reduced the intensity of the purple color of the radical solution. In addition, this evaluation was recorded by measuring the absorbance values of the samples at the end of the reaction at 517 nm. In previous studies, the IC_50_ value was found to be 30.6 μg/mL for L-adrenaline [74], 34.9 μg/mL for curcumin [61], 20.0 mg/mL for silymarin [75], 6.96 μg/mL for resveratrol [57], 17.77 μg/mL for olivetol [59], 77.00 μg/mL for taxifolin [58], and 16.06 μg/mL for eugenol [56] as natural plant metabolites. ABTS and DPPH free radical scavenging abilities were extensively used to determine the radical scavenging activities of aqueous blends, beverages, extracts, and pure substances [76]. In a recent study, the IC_50_ value was calculated as 45 μM for secoisolariciresinol diglycoside. In this study, secoisolariciresinol demonstrated IC_50_ values of 30 and 80 μM against DPPH and ABTS radicals, respectively [47]. In another study, it was determined that the different thermal preparations and in vitro digestion significantly affected sesame’s lignan content and antioxidant activity in defatted sesame meal [77]. It also was reported that the lignan content in flaxseed increased significantly through biochemical reactions during germination [78]. Zeng and coworkers isolated a new lignan, acutissimanide, with antioxidant activity from the bark of *Quercus acutissima*. The IC_50_ value of acutissimanide was calculated as 41.6 μM against DPPH radicals [79]. Similarly, a novel lignan glycoside, sagitiside A, together with known lignans of (+)-lyoniresinol-2α-O-β-D-glucopyranoside and (+)-5′-methoxyisolariciresinol-3α-O-β-D-glucopyranoside, demonstrated IC_50_ values of 55, 75, and 80 μM against DPPH radicals [80]. The flaxseed lignan secoisolariciresinol diglycoside and mammalian lignans enterodiol and enterolactone also were previously shown to be effective antioxidants against DNA damage and lipid peroxidation [81]. In another study, five lignan glycosides, including nudiposide, lyoniside, 5-methoxy-9-β-xylopyranosyl-(-)-isolariciresinol, icariside E3, and schizandriside, were isolated from a methanol extract of *Saraca asoca*. The IC_50_ values of these lignans varied between 30 and 104 μM for DPPH radical scavenging [82]. A new furofuran lignan, named petaslignolide A, was isolated from the methanolic extract of *Petasites japonicus* leaves. Its IC_50_ value was 113.04 μg/mL for DPPH radical scavenging [83].

## 4. Materials and Methods

### 4.1. Chemicals

DPPH^•^ (2,2-diphenyl-1-picrylhydrazyl), BHA (butylated hydroxyanisole), ABTS^•+^ (2,2′-azino-bis(3-ethylbenzothiazoline-6-sulphonic acid), Trolox (6-hydroxy-2,5,7,8-tetramethylchroman-2-carboxylic acid), BHT (butylated hydroxytoluene), α-tocopherol ((2R)-2,5,7,8-tetramethyl-2-[(4R,8R)-4,8,12-trimethyltridecyl]-3,4-dihydro-2H-1-benzopyran-6-ol), nordihydroguaiaretic acid, (-)-secoisolariciresinol, secoisolariciresinol diglycoside, α-(-)-conidendrin, enterodiol and enterolactone were purchased commercially from Sigma-Aldrich GmbH, Steinheim, Germany. All other chemicals used were analytical grade and were obtained from either Sigma-Aldrich or Merck (Kenilworth, NJ, USA). All used phyto and mammalian lignans and standard antioxidants were solubilized and diluted in ethanol.

### 4.2. Fe^3+^ Reducing Ability Assays

The Fe^3+^ reducing ability of phyto and mammalian lignans was determined according to the method of Oyaizu [84] as described previously [85]. For this purpose, different concentrations of phyto and mammalian lignans and 2.5 mL of K_3_Fe(CN) solution (1%), which was prepared in phosphate buffer (0.2 M, 2.5 mL, and pH 6.6) was incubated in the dark (50 °C, 30 min). Then, the same volume of TCA (10%) was transferred to the reaction medium. Next, an aliquot of FeCl_3_ (0.1%, 0.5 mL) was added, and the absorbance was measured at 700 nm [86].

### 4.3. Cu^2+^ Reduction Ability

The Cu^2+^ reduction activity of phyto and mammalian lignans was determined using a slight modification of the method of Apak et al. [87] as described in detail in [88]. Diverse concentrations of phyto and mammalian lignans were prepared and added to a CuCl_2_ solution (250 µL, 10 mM). Then, 0.25 mL of neocuproine solution was dissolved in ethanol (7.5 mM), and 0.25 mL of acetate buffer (1.0 M) was transferred. After 30 min, the absorbances were spectrophotometrically recorded at 450 nm.

### 4.4. The FRAP Reduction Ability

The FRAP reduction ability of phyto and mammalian lignans was determined according to a previous study [89]. The stock solution was used in this assay. First, different concentrations (10–30 μg/mL) of phyto and mammalian lignans and standard solutions were prepared in test tubes. Then, their volumes were supplemented with buffer solution (0.5 mL) and FeCl_3_ solution (2250 μL, 20 mM). Next, an equal volume of FRAP reagent was transferred, vortexed, and recorded at 593 nm after a short incubation period (10 min).

### 4.5. DPPH Radical Scavenging Activities

DPPH∙ scavenging activity of phyto and mammalian lignans was determined according to the Blois method [72] as described previously [90]. Stock solutions (1.0 mM) were added to the test tubes with different concentrations of phyto and mammalian lignans (10–30 μg/μL). Next, a 3 mL total volume was completed with ethanol. Finally, 1 mL of DPPH radical solution was transferred to each test tube, and incubated then the absorbance was measured at 517 nm.

### 4.6. ABTS Radical Scavenging Activities

Another radical removal method used to determine the ABTS^•+^ scavenging ability of phyto and mammalian lignans was developed according to a previous study [91]. Primarily, an ABTS solution (7.0 mM) was produced by adding K_2_S_2_O_8_ (2.45 mM), and the absorbance was adjusted to 0.700 ± 0.050 nm with buffer solution (0.1 M and pH 7.4) at 734 nm. Then, an aliquot of ABTS radicals (1 mL) was transferred to diverse concentrations of phyto and mammalian lignans (10–30 µg/mL).

### 4.7. IC_50_ Values Determination

For both radical scavenging assays, in order to calculate the IC_50_ values, first of all, three different lignan concentrations were used against radicals, and the corresponding absorbance values were obtained. Then, these values were converted into graphs. Finally, IC_50_ values were calculated from the equations obtained from these graphs [92,93,94].

### 4.8. Statistical Analysis

The experiments related to the antioxidant activities of phyto and mammalian lignans were performed in triplicate. The data were recorded as mean ± standard deviation and analyzed by SPSS (IBM SPSS 20). One-way analysis of variance (ANOVA) was realized by following the procedures. Significant differences between means were determined by Duncan’s multiple range tests; *p* < 0.05 was regarded as significant and *p* < 0.01 as very significant. The coefficient of multiple determination (r^2^) is a common output from computer regression packages and widely used in regression analysis. The r^2^ value represents the rate of variation in the response explained by the regression model.

## 5. Conclusions

This study demonstrated that phyto and mammalian lignans had considerable antioxidant and antiradical effectiveness in different in vitro bioanalytical methods when compared to the standard compounds of BHA, BHT, α-tocopherol, and Trolox. In this study, it was observed that phyto and mammalian lignans were active phenolic compounds that removed ROS and free radicals by donating an electron or hydrogen atom. Moreover, they were reported to have a large spectrum of potent biological activities. In addition, phyto lignans had powerful antiradical and antioxidant activities in different in vitro bioanalytical assays, including Fe^3+^-TPTZ, Fe^3+^, and Cu^2+^ reducing DPPH and ABTS^•+^ scavenging abilities when compared to mammalian lignans. In addition, the results showed that phyto and mammalian lignans possessed such activities in a concentration-dependent manner. They can be used effectively and safely in food or pharmaceutical products to minimize or prevent lipid peroxidation, delay the formation of some toxic oxidation products such as cyclic endoperoxides, maintain nutritional quality, and extend the shelf life of pharmaceutical and foodstuffs.

## Figures and Tables

**Figure 1 molecules-26-07099-f001:**
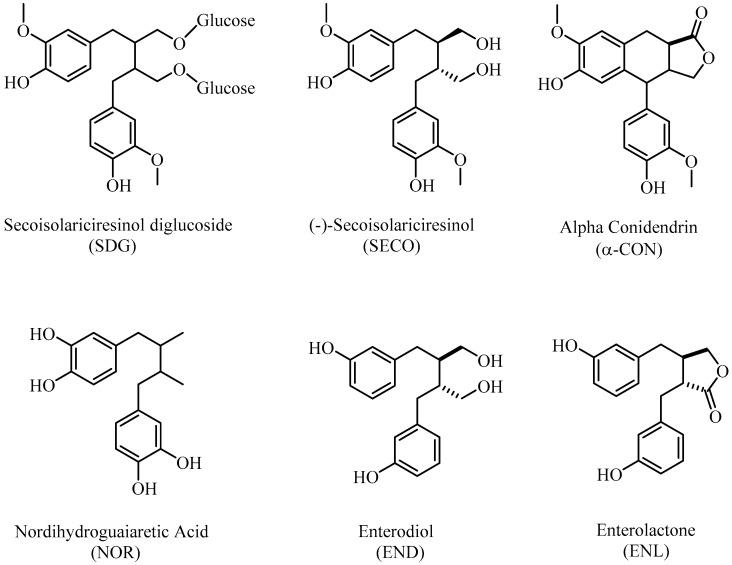
Chemical structures of lignans used in the present study.

**Figure 2 molecules-26-07099-f002:**
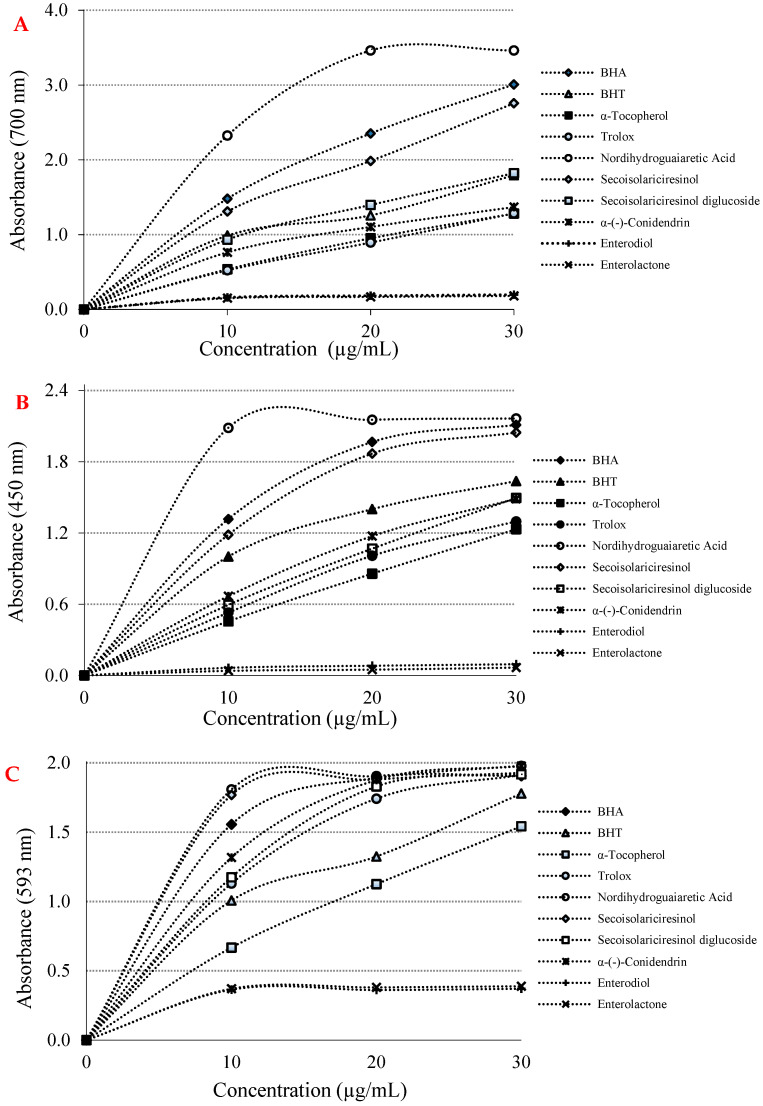
(**A**) Fe^3+^ reductive potential of lignans and references; (**B**) Cu^2+^ reducing ability of lignans and references; (**C**) Fe^3+^-TPTZ reducing ability of lignans and references.

**Figure 3 molecules-26-07099-f003:**
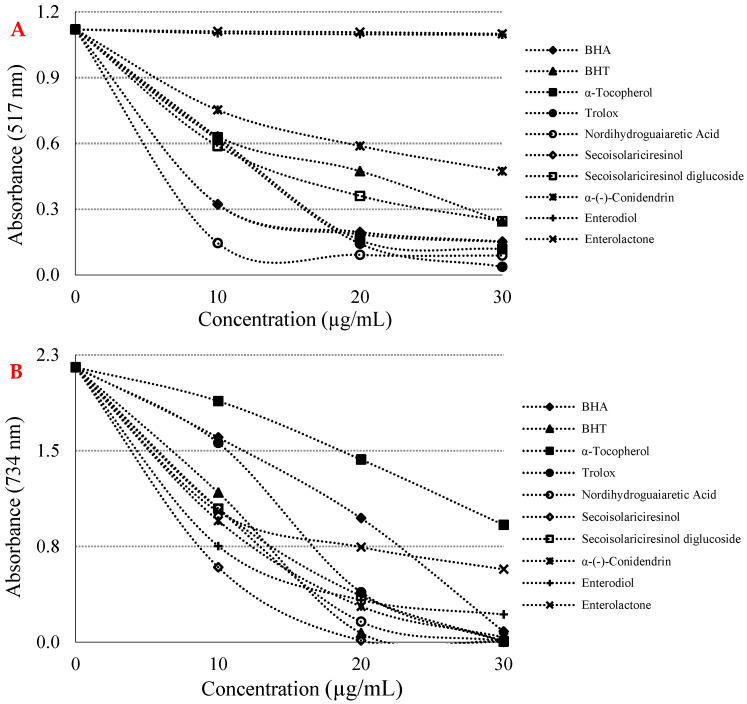
(**A**) 1,1-Diphenyl-2-picryl-hydrazyl (DPPH) free radical scavenging activity of lignans and references. (**B**) 2,2′-Azino-bis(3-ethylbenzthiazoline-6-sulfonic acid) (ABTS) radical scavenging activity of some phyto and mammalian lignans and standards.

**Figure 4 molecules-26-07099-f004:**
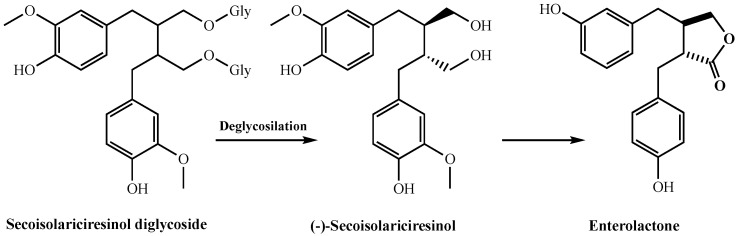
The metabolic pathway of secoisolariciresinol diglycoside in mammalian systems.

**Table 1 molecules-26-07099-t001:** Determination of reducing powers of the same concentration (10 µg/mL) of some phyto and mammalian lignans and standards by ferric ion (Fe^3+^), Fe^3+^-TPTZ, and cupric ion (Cu^2+^) reducing capacities.

Antioxidants	Fe^3+^ Reducing		Cu^2+^ Reducing		Fe^3+^-TPTZ Reducing	
	λ_700_	r^2^	λ_450_	r^2^	λ_593_	r^2^
BHA	1.480 ± 0.001 ^a^	0.9505	1.320 ± 0.002 ^a^	0.8290	1.560 ± 0.002 ^a^	0.6352
BHT	0.990 ± 0.00 ^a^	0.9198	1.000 ± 0.002 ^a^	0.8589	1.010 ± 0.002 ^a^	0.9085
Trolox	0.520 ± 0.002 ^a^	0.9907	0.530 ± 0.003 ^a^	0.9799	1.130 ± 0.003 ^a^	0.8600
α-Tocopherol	0.540 ± 0.003 ^a^	0.9841	0.460 ± 0.001 ^a^	0.9971	0.670 ± 0.001 ^a^	0.9815
Nordihydroguaiaretic acid	2.320 ± 0.003 ^a^	0.7704	2.090 ± 0.003 ^a^	0.4703	1.810 ± 0.003 ^a^	0.5289
Secoisolariciresinol	1.310 ± 0.001 ^a^	0.9655	1.180 ± 0.005 ^a^	0.8665	1.770 ± 0.003 ^a^	0.5248
Secoisolariciresinol diglycoside	0.930 ± 0.001 ^a^	0.9425	0.590 ± 0.002 ^a^	0.9917	1.170 ± 0.002 ^a^	0.8301
α-(-)-Conidendrin	0.760 ± 0.002 ^a^	0.9086	0.670 ± 0.002 ^a^	0.9666	1.320 ± 0.002 ^a^	0.7906
Enterodiol	0.160 ± 0.002 ^b^	0.6270	0.060 ± 0.002 ^b^	0.8867	0.360 ± 0.001 ^a^	0.4483
Enterolactone	0.150 ± 0.001 ^b^	0.6511	0.040 ± 0.001 ^b^	0.7938	0.370 ± 0.002 ^a^	0.4823

Superscript a is used to show very significant differences between parameters within each group and control value (*p* < 0.01). Superscript b shows significant differences (*p* < 0.5) between parameters within each group and control value.

**Table 2 molecules-26-07099-t002:** IC_50_ (μg/mL) values for DPPH^•^ and ABTS^•+^ scavenging activities of some phyto and mammalian lignans and standard antioxidants.

Antioxidants	DPPH^•^ Scavenging		ABTS^•+^ Scavenging	
	IC_50_	r^2^	IC_50_	r^2^
BHA	8.886	0.8406	16.552	0.9837
BHT	17.778	0.9397	13.007	0.9116
Trolox	6.478	0.9624	14.264	0.9660
α-Tocopherol	8.864	0.9437	27.829	0.9816
Nordihydroguaiaretic acid	6.601	0.8498	13.070	0.9174
(-)-Secoisolariciresinol	14.141	0.8426	12.252	0.8001
Secoisolariciresinol diglycoside	16.970	0.8998	13.547	0.9467
α-(-)-Conidendrin	23.295	0.9281	13.345	0.9140
Enterodiol	770.164	0.7746	13.878	0.8163
Enterolactone	932.167	0.9792	14.146	0.9070

## Data Availability

Data are available in a publicly accessible repository.

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
