# Peer review of "Evaluation of the Antioxidant and Antiradical Properties of Some Phyto and Mammalian Lignans"

_molecules, 2021, doi:10.3390/molecules26237099_

Round 1

Reviewer 1 Report

In this work the methods described by the Authors allows to determine the antioxidant and antiradical properties of some phyto and mammalian lignans.. Secondary metabolites have been analyzed by 2,2'-azino-bis (3-ethylbenzothiazoline-6-sulphonic acid) (ABTS•+) and 2,2- 12 diphenyl-1-picrylhydrazyl (DPPH·) radical scavenging assays. Additionally, the reducing ability of lignans were evaluated by Cu2+ reducing (CUPRAC), Fe3+ and [Fe3+-(TPTZ)2]3+ reducing (FRAP) abilities. Some phyto and mammalian lignans in plants showed considerable high antioxidant activity.

However, analyzed compounds are difficult to isolated and separated on individual compounds. In my opinion analyzed compounds have not been isolated and separated. Authors did not describe the origin of these compounds. This point need to be clarified by the authors. Please add a detailed description in point “Materials and Methods”.

Line 310, Author wrote…… “Both radicals scavenging assays were computed using the following equation: RSE (%) =[1-(As/Ac)] x 100 where RSE is radical scavenging influences, Ac is the absorbance amount of control and As is absorbance amount of sample [89-91].”

Information are not very clear because in table 2 and “results” there is no mention of RES, only IC50. This point need to be clarified by the authors.

Please check the Line 17: Enterolactone – lowercase; Line 93: ([34-36 - remove the parenthesis; Line 108 punctuation.Line 306: change K2S2O8 (2.45 nM) for K2S2O8 (2.45 mM) in accordance with the methodology.

Author Response

RESPONSES TO REVIEWER-1

-In this work the methods described by the Authors allows to determine the antioxidant and antiradical properties of some phyto and mammalian lignans. Secondary metabolites have been analyzed by 2,2'-azino-bis (3-ethylbenzothiazoline-6-sulphonic acid) (ABTS•+) and 2,2-diphenyl-1-picrylhydrazyl (DPPH·) radical scavenging assays. Additionally, the reducing ability of lignans were evaluated by Cu2+ reducing (CUPRAC), Fe3+ and [Fe3+-(TPTZ)2]3+ reducing (FRAP) abilities. Some phyto and mammalian lignans in plants showed considerable high antioxidant activity.

However, analyzed compounds are difficult to isolated and separated on individual compounds. In my opinion analyzed compounds have not been isolated and separated. Authors did not describe the origin of these compounds. This point need to be clarified by the authors. Please add a detailed description in point “Materials and Methods”.

RESPONSE: Many thanks to the reviewer due to his/her positive opinion. The lignans used in this study were not purified and isolated from any plant or animal source. All lignans used in this study were purchased from Sigma-Aldrich company. This information has been forgotten in the Materials and Methods section. We gave the indicated information "Chemicals" subhead as reviewer's recommendation.

-Line 310, Author wrote…… “Both radicals scavenging assays were computed using the following equation: RSE (%) =[1-(As/Ac)] x 100 where RSE is radical scavenging influences, Ac is the absorbance amount of control and As is absorbance amount of sample [89-91].”

Information are not very clear because in table 2 and “results” there is no mention of RES, only IC50. This point need to be clarified by the authors.

RESPONSE: The reviewer is right. There has been a mistake here. The information of “Both radicals scavenging assays were computed using the following equation: RSE (%) =[1-(As/Ac)] x 100 where RSE is radical scavenging influences, Ac is the absorbance amount of control and As is absorbance amount of sample [89-91].” was corrected as “For both radicals scavenging assays, in order to calculate the IC50 values, first of all, three different lignan concentrations were used against radicals and the corresponding absorbance values were obtained. Then, these values were converted into graphs. Finally, IC50values were calculated from the equations obtained from these graphs [89-91].”.

-Please check the Line 17: Enterolactone – lowercase; Line 93: ([34-36 - remove the parenthesis; Line 108 punctuation. Line 306: change K2S2O8 (2.45 nM) for K2S2O8 (2.45 mM) in accordance with the methodology.

RESPONSE:

Line 17            : “Enterolactone” was corrected as “enterolactone”.

Line 93            : “([34-36]” was corrected as “[34-36]”.

Line 108           : This sentence was corrected as “To this end the reducing capabilities of phyto and mammalian lignans were realized by three distinct and different reducing power methods including Fe3+(CN-)6 reduction, Fe3+-TPTZ reduction and Cu2+ ions reduction capacities.”.

Line 306           : “K2S2O8 (2.45 nM)” was corrected as “K2S2O8 (2.45 mM)”.

Reviewer 2 Report

The manuscript titled “Evaluation of the Antioxidant and Antiradical Properties of Some Phyto and Mammalian Lignans” by Leyla Polat Kose and Ä°lhami Gulcin, address an interesting issue concerning the antioxidant and antiradical properties of lignans. However, the manuscript presents several concerns that should be addressed.

The authors of this manuscript should revise and correct the grammar, syntax and use of words to present clarity of the studies and investigation before publication.

The authors should review the abstract as it is unclear. The abstract should contain the context in which the purpose of the research was placed, the results clearly described, and the conclusion and interpretation of the study.

DPPH • and ABTS • + scavenging activity of lignans and standard antioxidants and relative IC50 (μg/mL) values are reported in Figure 2 and Table 2, but the data seem inconsistent. The authors should clarify this discrepancy.

In the Materials and Methods section, it is reported that phyto and mammalian lignans were tested at different concentrations (10 - 30 μg/mL), but IC50 values > 30 μg/mL for DPPH • and ABTS • + scavenging activity of some lignans (including Enterodiol and Enterolactone) are reported. The authors should better specify the concentrations tested.

In Abstract (Line 17-18) it is reported “On the other hand, enterodiol and Enterolactone that is phenolic compounds has been observed very little antioxidant activity” and in Conclusions (Line 323-325) it is reported “The study demonstrated that phyto and mammalian lignans had considerable antioxidant and antiradical effectiveness in different in vitro bioanalytical methods when compared to the standards”. What does considerable mean?

In the Discussion section, the authors should report the literature data relating to the antioxidant and antiradical activity of phyto and mammalian lignans and should discuss them with the data shown in the present study.

The authors should add a paragraph on phyto lignan intakes and bioavailability of mammalian lignans.

In general, the citations section is too broad. Please shorten the number of citations and give preference to newer ones.

Please indicate significantly different values in Figures and Tables.

Please specify how phyto and mammalian lignans and standard antioxidants were solubilized and diluted.

Please correct trolox and tocopherol in Figure 1 and 2.

Please use different indicators for BHA and Secoisoriciresinol in Figure 1 C.

Author Response

RESPONSES TO REVIEWER-2:

The manuscript titled “Evaluation of the Antioxidant and Antiradical Properties of Some Phyto and Mammalian Lignans” by Leyla Polat Kose and Ä°lhami Gulcin, address an interesting issue concerning the antioxidant and antiradical properties of lignans. However, the manuscript presents several concerns that should be addressed.

RESPONSE: Many thanks to the reviewer due to his/her positive opinion. The recommended corrections have been made.

-The authors of this manuscript should revise and correct the grammar, syntax and use of words to present clarity of the studies and investigation before publication.

RESPONSE: The language of the article was checked by an expert as requested by the reviewer. The extensive corrections were made and labelled as red in the revised article.

- The authors should review the abstract as it is unclear. The abstract should contain the context in which the purpose of the research was placed, the results clearly described, and the conclusion and interpretation of the study.

RESPONSE: The summary was re-evaluated and the indicated corrections were made.

-DPPH and ABTS•+ scavenging activity of lignans and standard antioxidants and relative IC50 (μg/mL) values are reported in Figure 2 and Table 2, but the data seem inconsistent. The authors should clarify this discrepancy.

RESPONSE: Reviewer is right. In the DPPH and ABTS•+ scavenging activity of lignans, there were some errors in the IC50 calculations. Although the ranking did not change, the IC50 were calculated lower. These errors have been corrected in the relevant table and text.

- In the Materials and Methods section, it is reported that phyto and mammalian lignans were tested at different concentrations (10-30 μg/mL), but IC50 values > 30 μg/mL for DPPH and ABTS•+ scavenging activity of some lignans (including Enterodiol and Enterolactone) are reported. The authors should better specify the concentrations tested.

RESPONSE: The calculations to IC50 values were made again as stated above. Also, the sentence of “On the other hand, enterodiol and Enterolactone that is phenolic compounds has been observed very little antioxidant activity” was arranged as “Also, it was observed that enterodiol and enterolactone exhibited relatively weaker antioxidant activities when compared to phyto lignans or standard antioxidants including butylated hydroxytoluene (BHT), butylated hydroxyanisole (BHA), Trolox and α-Tocopherol”. Also, the sentence of “The study demonstrated that phyto and mammalian lignans had considerable antioxidant and antiradical effectiveness in different in vitro bioanalytical methods when compared to the standards” was arranged as “Also, phyto lignans had powerful antiradical and antioxidant activities in different in vitro bioanalytical assays, including Fe3+-TPTZ, Fe3+, and Cu2+ reducing, DPPH· and ABTS·+ scavenging abilities when compared mammalian lignans.”.

-In the Discussion section, the authors should report the literature data relating to the antioxidant and antiradical activity of phyto and mammalian lignans and should discuss them with the data shown in the present study.

RESPONSE: We gave some report the literature data relating to the antioxidant and antiradical activity of phyto and mammalian lignans and should discuss them with the data shown in the present study. For this purpose, the following information and related references were given in the discussion section:

“In a recent study, IC50 value was calculated as 45 μM for secoisolariciresinol diglycoside. In this study, secoisolariciresinol demonstrated IC50value of 30 and 80 μM against DPPH and ABTS radicals, respectively [47]. In another study, it was determined that the different thermal preparations and in vitro digestion significantly affected the sesame lignans content and antioxidant activity in defatted-sesame meal [77]. It has also been reported that the lignan content in flaxseed increases significantly through biochemical reactions during germination [78]. Zeng and coworkers isolated a new lignan, acutissimanide, with antioxidant activity from the bark of Quercus acutissima. IC50 value of acutissimanide was calculated as 41.6 μM against DPPH radicals [79]. Similarly, a novel lignan glycoside, sagitiside A, together with known lignans of (+)-lyoniresinol-2α-O-β-D-glucopyranoside and (+)-5’-methoxyisolariciresinol-3α-O-β-D-glucopyranoside demonstrated IC50 values of 55, 75 and 80 μM against DPPH radicals [80]. Also, the flaxseed lignan secoisolariciresinol diglycoside and mammalian lignans enterodiol and enterolactone were previously shown to be effective antioxidants against DNA damage and lipid peroxidation [81]. In another study, five lignan glycosides including nudiposide, lyoniside, 5-methoxy-9-β-xylopyranosyl-(-)-isolariciresinol, icariside E3, and schizandriside were isolated from a methanol extract of Saraca asoca. IC50 values of these lignans varies between 30-104 μM for DPPH radical scavenging [82]. A new furofuran lignan named as petaslignolide A was isolated from the methanolic extract of the of Petasites japonicus leaves. It’s IC50 values of 113.04 μg/mL for DPPH radical scavenging [83].”

-The authors should add a paragraph on phyto lignan intakes and bioavailability of mammalian lignans.

RESPONSE: We gave some information on phyto lignan intakes and bioavailability of mammalian lignans. For this purpose, the following information and related references were given in the discussion section:

“Phytolignans as secondary metabolites are widely distributed in grains, fruits, cereals and vegetables [47]. They are bioactive compounds with a wide scale of biological activities including antitumor, antioxidant and weakly estrogen-linked proprieties. There is great interest in promoting the inclusion of phytolignans-rich foods in human diets because of their potential human health benefits including the prevention of hypercholesterolemia, cardiovascular disease, breast and prostate cancers, osteoporosis and menopausal symptoms [48]. It was reported that humans rely on gut microbes to convert plant lignans into mammalian lignans. Intestinal bacteria metabolize plant lignans by processes such as glycosylation, demethylation, methylenation, dehydroxylation, ring cleavage and oxidation [49]. Several human intestinal bacteria involved in these processes have been isolated and characterized. Also, it is known that phytolignans are metabolized to mammalian lignans by both fecal and ruminal microbiota (Figure 2). Recent studies have shown that lignan metabolism in ruminant animals mainly occurs in the rumen, where plant lignans are converted to mammalian lignans by rumen microbes [50]. Plant lignans are converted to mammalian lignans by the gut microbiota and have three metabolic fates: First, they are excreted directly with the feces. Second, they are absorbed by the epithelial cells lining the colon, and are conjugated with glucuronic acid or sulfate. Third, they can be absorbed from the gut in their deconjugated form, and released into the bloodstream as conjugated free forms and reach the liver. Eventually, the conjugated mammalian lignans are excreted into physiological fluids including plasma and urine or returned to the gut via the enterohepatic circulation [49]. Although lignans and their biogenetic precursors have various health benefits, their application as antioxidants in the food industry is limited due to the poor liposolubility of such phenolic compounds. However, due to the positive and potential health benefits of phyto lignans, their usage as functional foods or as beneficial food additives is of great interest [47]. The antioxidant abilities of the secoisolariciresinol diglycoside as glycosylated lignan were recently studied. Also, secoisolariciresinol diglycoside is deglycosylated by microorganisms in the gut and forms secoisolariciresinol which is, in turn, converted into enterolactone (Figure 2) [51]. The presence of two hydrophilic glucose residues along with two phenol groups in the structure of secoisolariciresinol diglycoside is certain to reduce its liposolubility. The use of natural antioxidants like lignans in the food industry is an important issue to consider. It is very important that they are dissolved in lipid systems. Where these properties are insufficient, chemical modifications can sometimes solve problems.

-In general, the citations section is too broad. Please shorten the number of citations and give preference to newer ones.

RESPONSE: Some references were removed in line with the recommendations of the referees. For this purpose, 13 references were extracted. As you suggested above, when information on intakes and bioavailability of mammalian lignans was added, 6 references were added for this information. In addition, 7 references were added in the discussion section for comparison the antioxidant properties of lignans in the literature. In total, 13 references were added to the revised article and the reference number of the revised article increased to 94 again.

-Please indicate significantly different values in Figures and Tables.

RESPONSE: Different values were tried to be given in the figures, but the figures were very confused and intertwined. Therefore, giving different values to the figures was abandoned. So, the different values were given only in the Table 1.

-Please specify how phyto and mammalian lignans and standard antioxidants were solubilized and diluted.

RESPONSE: All used phyto and mammalian lignans and standard antioxidants were solubilized and diluted in ethanol. This sentence was written in section of “4.1. Chemicals”.

-Please correct trolox and tocopherol in Figure 1 and 2.

RESPONSE: Trolox and a-tocopherol were corrected in all figures.

-Please use different indicators for BHA and Secoisoriciresinol in Figure 1 C.

RESPONSE: Figure 1 was corrected as Figure 2. Figures were renumbered. Also, different symbols were given for BHA and Secoisoriciresinol in Figure 2C.

Round 2

Reviewer 1 Report

I appreciate the efforts put in by the authors to address all my concerns. I feel that the manuscript is now acceptable for its possible publication.